# Effect of ATR Inhibition in RT Response of HPV-Negative and HPV-Positive Head and Neck Cancers

**DOI:** 10.3390/ijms22041504

**Published:** 2021-02-03

**Authors:** Rüveyda Dok, Mary Glorieux, Marieke Bamps, Sandra Nuyts

**Affiliations:** 1Laboratory of Experimental Radiotherapy, Department of Oncology, KU Leuven, University of Leuven, 3000 Leuven, Belgium; ruveyda.dok@kuleuven.be (R.D.); maryglorieux@hotmail.com (M.G.); marieke.bamps@kuleuven.be (M.B.); 2Department of Radiation Oncology, Leuven Cancer Institute, UZ Leuven, 3000 Leuven, Belgium

**Keywords:** head and neck cancers, human papillomavirus, DNA damage response, radiotherapy, ATR inhibition

## Abstract

Radiotherapy (RT) has a central role in head and neck squamous cell carcinoma (HNSCC) treatment. Targeted therapies modulating DNA damage response (DDR) and more specific cell cycle checkpoints can improve the radiotherapeutic response. Here, we assessed the influence of ataxia-telangiectasia mutated and Rad3-related (ATR) inhibition with the ATR inhibitor AZD6738 on RT response in both human papillomavirus (HPV)-negative and HPV-positive HNSCC. We found that ATR inhibition enhanced RT response in HPV-negative and HPV-positive cell lines independent of HPV status. The radiosensitizing effect of AZD6738 was correlated with checkpoint kinase 1 (CHK1)-mediated abrogation of G2/M-arrest. This resulted in the inhibition of RT-induced DNA repair and in an increase in the percentage of micronucleated cells. We validated the enhanced RT response in HPV-negative and HPV-positive xenograft models. These data demonstrate the potential use of ATR inhibition in combination with RT as a treatment option for both HPV-negative and HPV-positive HNSCC patients.

## 1. Introduction

Head and neck squamous cell carcinoma (HNSCC) are subdivided in human papillomavirus (HPV)-positive and HPV-negative HNSCC. The latter is most often associated with alcohol and tobacco use [1,2]. Radiotherapy (RT) alone or in combination with chemotherapy is the major treatment form for both HPV-negative and HPV-positive locally advanced HNSCC. Despite considerable technological advances in RT and new insights in the biological interaction between RT and the tumor, many HNSCC patients experience local tumor recurrences. This is especially the case for the HPV-negative group of HNSCC patients, highlighting the differential response to RT between the two groups of HNSCC [1,2,3].

Deoxyribonucleic acid (DNA) DNA damage response (DDR) and DNA repair pathways form a prominent mode of radiation resistance and result in a reduction of radiotherapeutic efficacy [4]. Therefore, targeted therapies modulating the DDR and DNA repair capacity of cancer cells can improve the radiotherapeutic response [5]. Recently, we have shown that HPV-positive and HPV-negative HNSCC show differences but also similarities in RT-induced DDR, and these differences or similarities should be employed for novel radiosensitization approaches. One of the similarities was the importance of cell cycle-related genes in the RT response of HNSCC cells [6].

The cell cycle consists of checkpoints, which are activated in the presence of DNA damage, that are crucial for the survival of cells [7,8]. Ataxia-telangiectasia mutated (ATM) and ataxia-telangiectasia mutated and Rad3-related (ATR) are two important kinases that sense DNA damage whereupon they activate the cell cycle and DNA repair processes. ATR and ATM phosphorylate checkpoint kinase 1 (CHK1) and CHK2 respectively to eventually activate the G2/M cell cycle checkpoint, inducing cell cycle arrest and enabling DNA repair. Dysfunction of the G1 cell cycle checkpoint, often via defects in the p53-Rb pathways, is one the major characteristics of cancer cells. Hence, cancer cells rely on the S and G2/M phase checkpoints for cell cycle arrest and DNA repair after damage [9]. Several studies have shown that ATR inhibition could increase the radiosensitization of cancer cells [10,11,12,13,14]. However, no profound investigation on the effect of ATR inhibition in different subgroups of HNSCC has been made. In this study, we will investigate the radiosensitizing effect of the ATR inhibitor AZD6738 both in vitro and in vivo models of HPV-positive and HPV-negative HNSCC.

## 2. Results

### 2.1. ATR Inhibition Induces Radiosensitization of HPV-Negative and HPV-Positive HNSCC In Vitro

To assess the radiosensitization potential of the ATR inhibition in different HNSCC subgroups, a clonogenic assay in three HPV-negative (SQD9, CAL27, and SCC61) and three HPV-positive (SCC154, SCC104, and SCC47) HNSCC cells was performed. AZD6738 as monotherapy significantly reduced the clonogenic growth compared to the vehicle treated control when administered at doses of 0.5 µM in SCC104 (average difference of 28%) and CAL27 (average difference 14%) cells (Figure 1).

The administration of AZD6738 in combination with RT resulted in decreased clonogenic cell survival in both HPV-negative and HPV-positive HNSCC cell lines (Figure 2). The degree of response to the combination treatment was variable depending on the cell line, the RT dose, and the drug dose, but not on the HPV status, as could be seen by the average dose enhancement factor (DEF) at 2Gy (Figure 2).

### 2.2. ATR Induces Abrogation of Cell Cycle Checkpoint by Inhibiting CHK1 Phosphorylation

Since radiosensitization by ATR inhibition has been related to the abrogation of radiation-induced G2/M cell cycle checkpoints, we assessed whether we could see this in HPV-negative and HPV-positive HNSCC cells. As expected, RT (6Gy) induced cell cycle arrest by G2/M accumulation in all HNSCC cells compared to the vehicle-treated control cells (Figure 3). The G2/M arrest correlated with the baseline radiosensitivity of the cell lines, which lasted for 24 h in all HPV-positive cells and the radiosensitive HPV-negative SCC61 cells (Figure 3). The addition of 0.5 µM of AZD6738 to RT abrogated RT-induced G2/M arrest. ATR inhibition with 0.5 µM of AZD6738 alone did not change the cell cycle distribution compared to the vehicle-treated conditions in both HPV-negative and HPV-positive HNSCC cells (Figure 3).

It is known that RT can activate the ATR–CHK1 axis to induce cell cycle arrest [13,15,16]. To assess the effect of ATR inhibition on RT-induced ATR-CHK1 pathway activation, we investigated the ATR-mediated CHK1 phosphorylation levels on ser345 (Figure 4). We verified that RT increased the phosphorylation of CHK1 at 4, 12, and 24 h after RT. However, cell line-dependent variation in the degree of activation could be seen (Figure 4). The addition of 0.5 µM of AZD6738 inhibited the phosphorylation of CHK1 both in monotherapy and in the combination treatment conditions (Figure 4), further confirming that ATR inhibition radiosensitized HNSCC cells through cell cycle checkpoint inhibition.

### 2.3. ATR Inhibition Induces an Increase in Residual γH2AX Levels and Micronuclei in HNSCC Cells

To investigate the effect of ATR inhibition on RT-induced DDR, we assessed the residual gamma histone 2AX (γH2AX) levels, which is a key event in the DDR response, 24 h after RT by flow cytometry (Figure 5). An average of 1.8-fold increase in γH2AX levels could be detected 4 h after RT (positive control samples) compared to the non-irradiated controls. In HPV-negative HNSCC cells, increased γH2AX levels were detected only in conditions in which ATR inhibition was combined with RT (Figure 5). In contrast, residual γH2AX levels were present for both the combination treatment and RT treatment condition in HPV-positive cells (Figure 5). The latter correlates with the defective DNA repair seen in HPV-positive cells [17,18,19,20,21,22,23,24,25]. Persistent RT-induced γH2AX levels make it difficult to make firm conclusions about the role of ATR inhibition in the generation of additional double-strand breaks (DSBs) in HPV-positive HNSCC cells. Of note, ATR inhibition alone did not result in increased residual γH2AX levels.

It is known that cell division with damaged DNA can result in mitotic catastrophe [10,26,27]. Since our data indicate that ATR inhibition results in cell cycle progression in the presence of DNA damage, we assessed whether we could see increased mitotic catastrophe by micronucleus assay in AZD6738-treated conditions (Figure 6, Appendix A). As expected, we could detect an increase in micronuclei in RT-treated conditions (Figure 6). The addition of an ATR inhibitor to RT resulted in an increase in micronuclei compared to RT in HPV-positive SCC104 and SCC154 cells. In HPV-positive SCC47 cells, no additional increase in RT-induced micronuclei was seen (Figure 6). In contrast, an inhibition of ATR resulted in an increase in micronuclei in all HPV-negative cells compared to the RT treated condition. Altogether, our results suggest that ATR inhibition induces cell cycle progression of cells with RT-induced DNA damage, resulting in an increase in micronuclei in HNSCC cells.

### 2.4. Preclinical Assessment of ATR Inhibitor AZD6738 as Radiosensitizer for HPV-Positive and HPV-Negative HNSCC

Next, we assessed the radiosensitization potential of ATR inhibitor AZD6738 using HPV-negative CAL27 and HPV-positive SCC154 xenograft models. In line with the *in vitro* data, ATR inhibition did result in an inhibition of RT-induced phosphorylation of CHK1 (Figure 7). When compared to the vehicle-treated controls, ATR inhibition alone significantly reduced the tumor volume of the HPV-negative CAL27 model from 22 days after the start of treatment (Figure 7). In contrast, AZD6738-treated HPV-positive SCC154 xenografts did not show a reduction in the tumor volumes compared to the vehicle-treated controls (Figure 7). Consistent with the *in vitro* data, we found that ATR inhibition enhanced the RT response in both HPV-negative CAL27 and HPV-positive SCC154 models (Figure 7). Although the absence of a clear tumor regrowth in RT-treated conditions of the HPV-negative xenografts hampers the possibility to assess the tumor growth delay, a limited but significant reduction in tumor volumes could be seen in xenografts treated with the combination of ATR inhibitor and RT (Figure 7). In the HPV-positive model, the combination treatment resulted in a tumor growth delay of 24 days compared to the RT-treated controls (Figure 7). Altogether, these results show the preclinical potential of combining ATR inhibitors with RT for both HPV-negative and HPV-positive HNSCC cells.

## 3. Discussion

The differential RT response between HPV-negative and HPV-positive HNSCC is at least partially attributed to defects in DDR [19,24]. In a previous study, we have shown that the inhibition of DDR regulators and cell cycle-related genes resulted in the radiosensitization of both groups [6]. Since ATR is crucial for maintaining the genomic stability, several potent, selective ATR inhibitors such as VX-970, AZD6738, and recently BAY 1895344 have been developed [28,29,30].

In addition, ATR is one of the major regulators of DDR, and numerous studies underline the role of ATR in the viral replication process of early HPV infections [31,32,33,34]. However, the role of ATR on DDR response in HPV-positive HNSCC remains understudied. In this study, we assessed the importance of ATR in RT-induced DDR response of both HPV-negative and HPV-positive HNSCC by inhibiting ATR with the selective ATR inhibitor AZD6738.

The radiosensitizing potential of ATR inhibitors in HPV-negative HNSCC has been demonstrated in numerous studies [10,14,35,36]. Recently, Vitti et al. assessed the radiosensitization potential of ATR inhibition in two HPV-positive (SCC47 and SCC090) and two HPV-negative HNSCC cell lines (SCC6 and SCC74A) [37]. They observed a lesser impact of DDR inhibitors in combination with RT in HPV-positive HNSCC cells compared to HPV-negative HNSCC cells [37]. In contrast to their report, we demonstrate that the degree of radiosensitization of the ATR inhibitor AZD6738 differed between different cell lines and was independent of HPV status. In line with our findings, ATR inhibition enhanced the sensitivity to cisplatin in a cell line-dependent manner in both HPV-positive and HPV-negative HNSCC [38]. This despite that defective DDR has been reported as a marker for ATR sensitivity [35,39,40] and differences in DDR have been implied as a potential mechanism for ATR-mediated sensitivity to DNA-damaging agents [38].

Although DSBs are the most lethal for cells, RT also produces single-strand breaks (SSBs), which if unrepaired could lead to DSBs [41]. In replicating cells, ATR is important for the prevention of fork collapse when encountering SSBs lesions [15,16] and therefore important for prevention of the conversion of these lesions to DSBs. In line with this, ATR inhibitors have been shown to stimulate mitosis with damaged DNA, thereby inducing mitotic catastrophe, which eventually can lead to cell death [10]. To assess whether we could detect this mechanism of action in HPV-positive and HPV-negative HNSCC, we opted for flow cytometry-based analysis of γH2AX levels with additional labeling with propidium iodide for cell cycle analysis. Detection of γH2AX has become a surrogate for DSBs; however, not all γH2AX foci identify DSBs [42,43]. Moreover, in contrast to fluorescent microscopy, flow cytometry-based analysis of γH2AX detects the fluorescence intensity measurement of individual cells, without differentiating whether the signal is from one focus, many foci, or from background intensity [44]. This can result in lower detection sensitivity compared to fluorescent microscopy and can explain the high background levels seen in our results. Nevertheless, our results show that the addition of AZD6738 to RT increased RT-induced residual γH2AX levels for HPV-negative cells and reduced RT-induced G2/M cell cycle arrest in all cell lines. Since HPV-positive cells show persistent γH2AX at 24 h in RT-treated conditions, the influence of ATR inhibition on RT-induced γH2AX levels is not clear. ATR inhibition alone has been reported to induce G1/S cell cycle arrest in conditions with a non-defective G1 cell cycle checkpoint [10,13,35]. Since most cancer cells are known to show dysfunction of the G1 checkpoints, ATR inhibition alone did not result in differences in cell cycle progression compared to the untreated non-irradiated control cells.

Next, we assessed the induction of micronuclei as a measure of ATR-mediated radiosensitivity. Our results showed that ATR inhibition stimulated cell cycle progression of cells with RT-induced DNA damage, resulting in an increased percentage of micronucleated cells for the majority of HNSCC cells. It should be noted that the highest fold increase was seen in CAL27 for the HPV-negative cells (2.3 fold) and in SCC154 for the HPV-positive cells (1.7 fold). In line with this, CAL27 and SCC154 showed the highest radiosensitization potential in our panel of tested cell lines.

Next, we assessed the preclinical potential of AZD6738 in combination with RT in xenograft models generated from CAL27 and SCC154 cells. In concordance with the *in vitro* data, a significant inhibitory effect of the ATR inhibitor on tumor growth was only seen in HPV-negative CAL27 xenografts. This corresponds with the limited but significant effect of AZD6738 treatment on clonogenic growth of CAL27 cells. ATR inhibition did enhance the RT response in both HPV-negative CAL27 and HPV-positive SCC154 models, but the effect of the combination treatment was limited in the former. Dunne et al. showed that the combination treatment did not increase RT-induced toxicity-like fibrosis [45]. In concordance, the combination treatment did not result in excessive weight loss or distress in our study-included xenografts.

The limited effect seen in HPV-negative xenografts and the absence of a clear mechanistic insight highlight the need for further assessment of the enhanced radiosensitivity in a larger panel of in vitro and in vivo HNSCC models with different HPV backgrounds. Moreover, ATR inhibition in combination with RT has been shown to modulate the immune tumor microenvironment, leading to immunologic memory and lasting antitumor immunity [46,47]. Therefore, the assessment of different treatment schedules are important and should be extended to immune competent mice.

Results from the PATRIOT trial (NCT02223923) proved that AZD6837 monotherapy is well tolerated in patients with solid tumors [48]. The monotherapy of AZD6738 resulted in confirmed partial response (7% of participants) and a high proportion of prolonged stable disease (22% of participants) [48]. A parallel dose-escalation study combining AZD6738 with palliative RT is ongoing.

Altogether, our results indicate that ATR inhibition radiosensitizes HNSCC independent of HPV status.

## 4. Materials and Methods

### 4.1. Cell Culture and Consumables

Three HPV-positive (UPCI-SCC-154, UM-SCC-104 and UM-SCC-47) and three HPV-negative (SQD9, SCC61 and CAL27) HNSCC cell lines were used as described previously [6]. All cell lines were authenticated by ATCC via short-tandem repeat profiling, and all experiments were performed with mycoplasma-free cells. AZD6738 was bought from Selleck Chemicals (S7693, Houston, TX, USA), and for in vitro testing, a stock solution of 10 mM was made in dimethylsulfoxide (DMSO) (D2438, Sigma-Aldrich, Saint Louis, MO, USA). For in vivo use, AZD6738 was dissolved in 10% DMSO, 40% propylene glycol (202398, Sigma-Aldrich, Saint Louis, MO, USA), and 50% sterile dH_2_O.

### 4.2. Colony Assay

Cells were seeded in 6-well plates, and after attachment overnight, cells were treated with 0.25 or 0.5 µM AZD6738. The chosen concentrations of AZD6738 are based on [10,49]. Two hours after drug exposure, cells were irradiated with RT 0-6Gy using a Baltograph (199 kV, 15 mA, Balteau NDT, Oupeye, Belgium) as previously described [6,50]. AZD6738 was replaced with fresh medium 24 h after drug exposure. After a 14–21 days incubation period, colonies were fixed as described earlier [6]. For the analysis, the plating efficiencies were determined by dividing the amount of counted colonies by the amount of seeded cells. For each treatment condition (0, 0.25, or 0.5 µM), the survival fractions were calculated by normalization to the plating efficiency of non-irradiated controls of the respective treatment conditions (0, 0.25, or 0.5 µM). The DEF at 2Gy (DEF2Gy) is defined as the ratio of survival at 2Gy of the control samples divided by the survival at 2Gy of the 0.25 µM or 0.5 µM-treated samples.

### 4.3. Western Blotting

For immunoblotting, cells were treated with 0.5 µM AZD6738 and/or 6Gy. Control conditions were treated with equal concentrations of DMSO. Two hours after drug exposure, cells were irradiated with 6Gy, and after 24 h, the drug was replaced with medium. At the indicated time points, cells were lysed with radioimmunoprecipitation assay (RIPA) buffer containing sodium vanadate phosphatase (S6508, Sigma-Aldrich, Saint Louis, MO, USA) and cOmplete™, EDTA-free Protease Inhibitor Cocktail (11873580001, Sigma-Aldrich, Saint Louis, MO, USA). For protein analysis of the xenograft tissue, tumors were snap frozen in liquid nitrogen at the evening of the last treatment day and stored at −80 °C. Tumor pieces were homogenized mechanically and lysed with RIPA buffer containing sodium vanadate phosphatase and cOmplete™, EDTA-free Protease Inhibitor Cocktail.

Protein concentrations were determined using the Bradford method (5000006, Bio-Rad, Hercules, CA, USA), and equal amounts of protein samples were separated on NUPAGE gels (NP0321, Invitrogen, Carlsbad, CA, USA). SeeBlueTM (LC5925, Invitrogen, Carlsbad, CA, USA) was used as protein standard. Proteins were transferred onto a polyvinylidene fluoride membrane (1620177, Bio-Rad, Hercules, CA, USA) and blocked with 5% non-fatty dry milk. Membranes were incubated overnight on 4 °C with primary antibodies for CHK1 (#2360, CST, Danvers, MA, USA), phospho-CHK1 ser345 (#2348, CST, Danvers, MA, USA), and vinculin (clone hVIN-1, V9131, Sigma-Aldrich, Saint Louis, MO, USA) was used as loading control. Visualization was performed with enhanced chemiluminescence reagent (ECL Ultra, NEL111001EA, Perkin Elmer, Waltham, MA, USA) using a chemiluminescence imager (AI680 westernblot imager, GE Healthcare, Chicago, IL, USA). Densitometry analysis was performed using ImageQuantTL software (GE Healthcare, Chicago, IL, USA). All densitometry values were normalized to the loading control. Relative values were calculated by dividing each condition by the control vehicle-treated condition.

### 4.4. Flow Cytometry

Cells were treated with 0.5 µM AZD6738 and/or 6Gy. For the combination treatment, the cells were irradiated 2 h after drug exposure. Twenty-four hours after drug exposure, the drug was replaced by medium. At the indicated time points, cells were fixed as described previously [6] and stained with 10 µg/mL propidium iodide (P4170, Sigma-Aldrich, Saint Louis, MO, USA) containing 100 µg/mL RNAse A (12091021, Invitrogen, Carlsbad, CA, USA) and H2AX antibody (Alexa Fluor 488 Mouse anti-H2AX pS139, 560445, BD Biosciences, San Jose, CA, USA). Cells were assessed by FACSverse (BD Biosciences, San Jose, CA, USA).

### 4.5. Micronuclei Assay

Cells were plated on µClear 96-well plates (655096, Greiner Bio-one, Vilvoorde, Belgium) and treated with 0.5 µM AZD6738 and/or 6Gy. For the combination treatment, the cells were irradiated 2 h after drug exposure. Samples were fixed as described in [6]. The nuclei were counterstained with DAPI (D9542, Sigma-Aldrich, Saint Louis, MO, USA). Immunofluorescence images were acquired with In Cell Analyzer 2000 (GE Healthcare, Chicago, IL, USA).

### 4.6. Xenograft Models

CAL27 cells (2 × 10^6^) or SCC154 cells (7.5 × 10^6^) were subcutaneously injected in each flank of 6–7-week-old female nu/nu Naval Medical Research Institute (NMRI) mice (Janvier Labs, France). The mice were either treated with vehicle or with AZD6738 (50 mg/kg body weight), which were all administered via gavage. AZD6738 was delivered 5 days before RT treatment and 2 h before fractionated RT (2Gy/fraction) with a total dose of 10Gy. Tumor volumes were determined with caliper measurements and calculated as V **=** π/6x_1xd_2xd_3. The bodyweight and health of the mice were monitored daily during treatment and three times per week during follow-up. The experiment was performed according to the Ethical committee Animal Experimentation of KU Leuven (P163/2017; approval date 29 September 2017).

### 4.7. Statistical Analysis

For in vitro and in vivo experiments, statistical analysis was performed with ANOVA using GraphPad software. All tests were considered statistically significant for *p* < 0.05.

## 5. Conclusions

In this study, we show that ATR inhibition enhanced RT response in HPV-negative and HPV-positive HNSCC in vitro and in vivo. We show that the radiosensitizing effect of AZD6738 correlates with CHK1-mediated abrogation of G2/M-arrest and subsequent progression of cells with RT-induced DNA damage and an increase in micronucleated cells. These data demonstrate the potential use of ATR inhibition in combination with RT as treatment option for both HPV-negative and HPV-positive HNSCC patients.

## Figures and Tables

**Figure 1 ijms-22-01504-f001:**
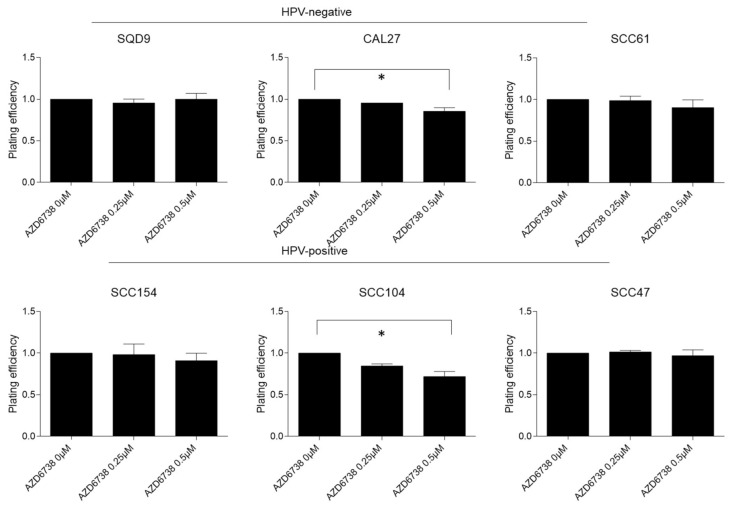
Effect of ATR inhibition on clonogenic survival in HPV-negative and HPV-positive HNSCC cells. HPV-negative (SQD9, CAL27, and SCC61) and HPV-positive (SCC154, SCC104, and SCC47) cells were treated for 24 h with DMSO or AZD6738 at indicated doses. Plating efficiency was determined by the means of clonogenic survival and is shown as the mean clonogenic survival fraction ± s.e.m. relative to vehicle treated control cells, *n* = 3. * *p*-values were calculated with one-way ANOVA with Bonferroni correction for multiple testing.

**Figure 2 ijms-22-01504-f002:**
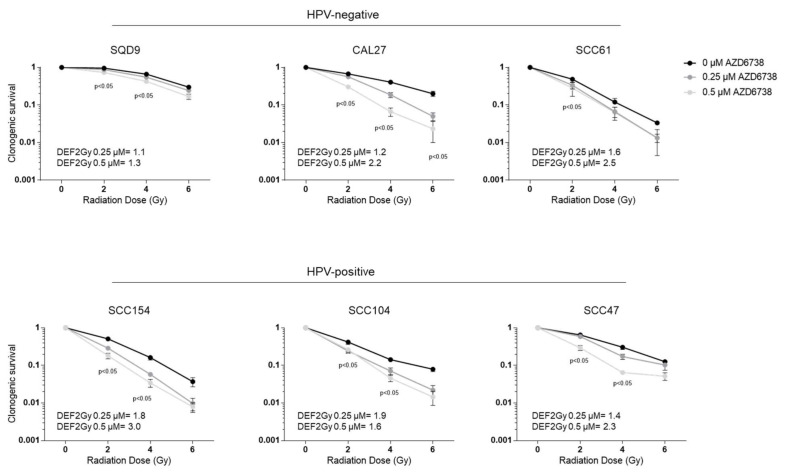
Effect of ATR inhibition on radiotherapy (RT) response of HPV-negative and HPV-positive head HNSCC cells. HPV-negative (SQD9, CAL27, and SCC61) and HPV-positive (SCC154, SCC104, and SCC47) cells pre-treated for 2 h with DMSO or AZD6738 were irradiated with the indicated RT doses. Clonogenic cell survival is shown as the mean clonogenic survival fraction ± s.e.m., *n* = 3. *P*-values were calculated with two-way ANOVA with Bonferroni correction for multiple testing. The dose enhancement factor at 2Gy (DEF2Gy) is defined as the ratio of average survival at 2Gy of the control samples divided by the average survival at 2Gy of 0.25 or 0.5 µM treated samples.

**Figure 3 ijms-22-01504-f003:**
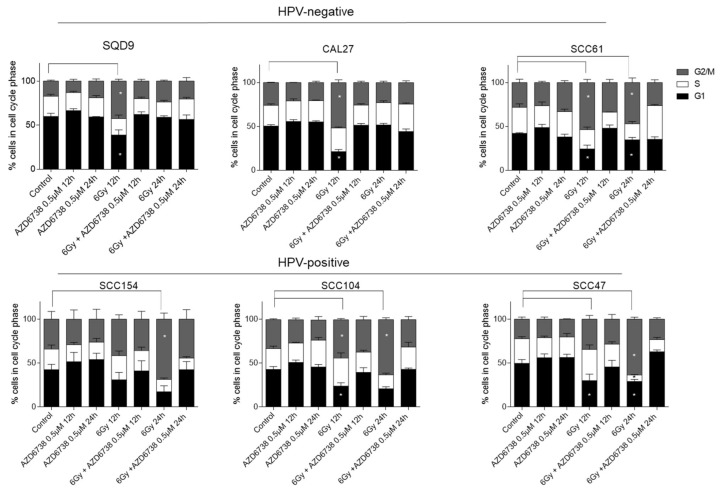
Effect of ATR inhibition on cell cycle progression. The percentage (%) of HPV-negative (SQD9, CAL27, and SCC61) and HPV-positive (SCC154, SCC104, and SCC47) HNSCC cells distributed in cell cycle phases after treatment with 0.5 µM AZD6738 and/or 6Gy of RT. In all the drug-treated conditions, the AZD6738 is incubated for 12 h and 24 h. For the combination treatment conditions, AZD6738 is given 2 h before RT and is removed after 12 and 24 h. The indicated time points on the x-axis indicate the time after drug and/or RT exposure. Data are represented as the mean ± SEM, *n* = 3. * *p*-values were calculated with two-way ANOVA with Tukey’s multiple correction.

**Figure 4 ijms-22-01504-f004:**
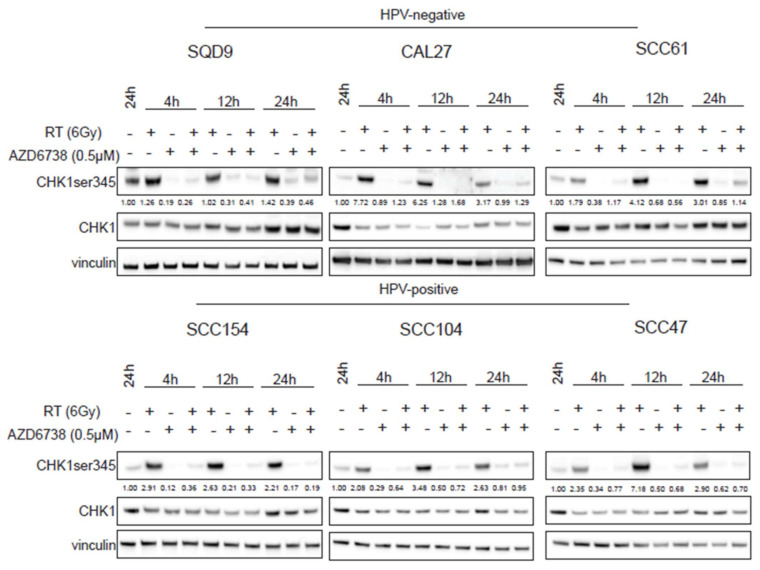
Effect of ATR inhibition on CHK1 activation. HPV-negative (SQD9, CAL27, and SCC61) and HPV-positive (SCC154, SCC104, and SCC47) HNSCC cells were treated with DMSO or AZD6738 (0.5 µM) 2 h before RT. At the indicated time points, total cell lysates were prepared to perform western blotting for CHK1 ser345 (51 kDa), CHK1 (51 kDa), and vinculin (97 kDa). Densitometry quantification was performed and relative expression levels were corrected to the loading control (vinculin). kDa = molecular weight as determined by the protein standard.

**Figure 5 ijms-22-01504-f005:**
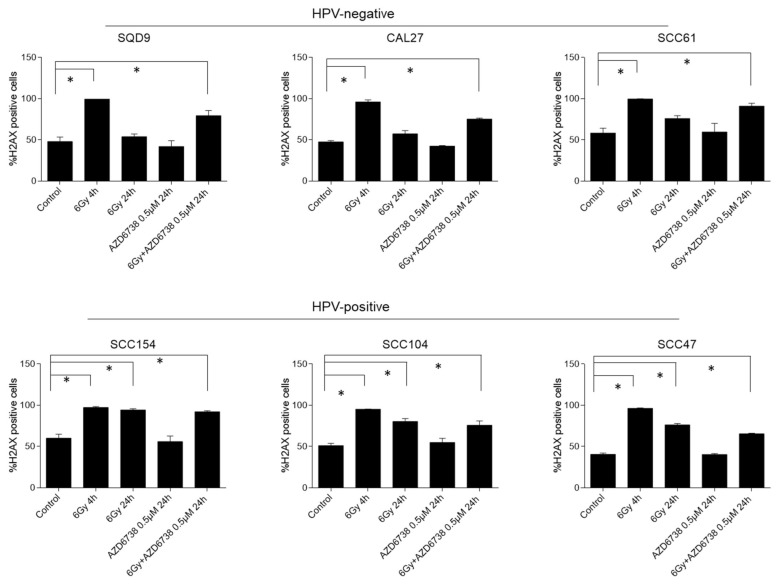
Effect of ATR inhibition on residual γH2AX levels. HPV-negative (SQD9, CAL27, and SCC61) and HPV-positive (SCC154, SCC104, and SCC47) HNSCC cells were exposed to AZD6738 (0 or 0.5 µM) and/or 6Gy of RT, and γH2AX levels were assessed 4 h (positive control) and 24 h after RT. Data are represented as the mean ± SEM, *n* = 3. * *p* values were determined with one-way ANOVA with Bonferroni correction for multiple testing.

**Figure 6 ijms-22-01504-f006:**
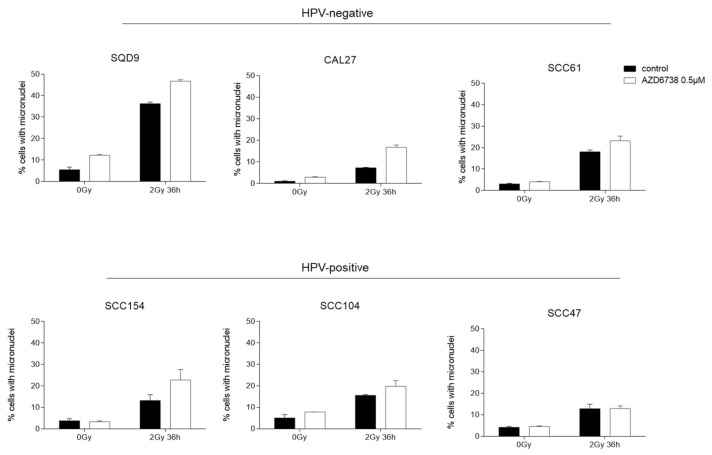
Effect of ATR inhibition on micronuclei formation. HPV-negative (SQD9, CAL27, and SCC61) and HPV-positive (SCC154, SCC104, and SCC47) HNSCC cells were exposed to 0.5 µM of AZD6738 and/or 6Gy of RT to assess the percentage of micronucleated cells to the total number of cells. Data are represented as the mean ± SEM, *n* = 2.

**Figure 7 ijms-22-01504-f007:**
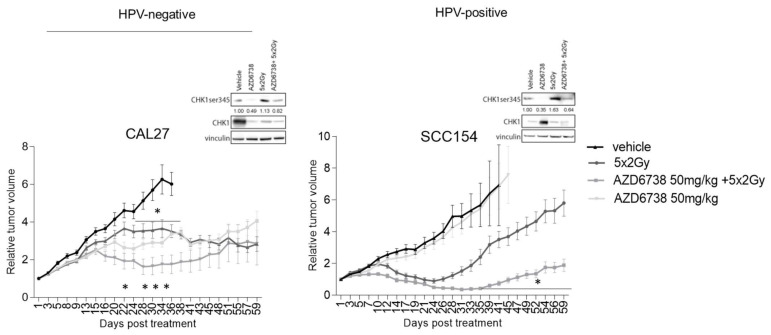
Effect of ATR inhibition on RT response of HPV-negative and HPV-positive HNSCC xenografts. Relative tumor volume curves of HPV-negative CAL27 and HPV-positive SCC154 xenografts treated with vehicle (10% DMSO/40% propylene glycol) or AZD6738 (50 mg/kg) with(out) RT (5 × 2Gy). Relative tumor growth curves were normalized to the tumor volume at the start of treatment Day 1. Day 1 indicates the start of treatment. Data are presented as mean ± s.e.m, *n* = 10. * *p*-value < 0.05 were determined by two-way ANOVA with Bonferroni correction for multiple testing. Western blot analysis of xenograft samples of the HPV-negative CAL27 model (left) and the HPV-positive model SCC154 (right) show the effect of ATR inhibition on CHK1 activation. Tumors for Western blot analysis were harvested at the evening of the last treatment day per schedule. CHK1ser345 (51 kDa), CHK1 (51 kDa), and vinculin (97 kDa) antibodies were used for immunoblotting. N = tumors; kDa = molecular weight as determined by the protein standard.

## Data Availability

Data underlying this article can be made available upon request.

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
