# Peer review of "Effect of ATR Inhibition in RT Response of HPV-Negative and HPV-Positive Head and Neck Cancers"

_ijms, 2021, doi:10.3390/ijms22041504_

Round 1
Reviewer 1 Report
This is a study on the use of ATR inhibitor, AZD6738, to improve response of radiotherapy in head and neck cancer. The study was based on in vitro and in vivo experiments. The authors reveal that the degree of radiosensitization by AZD6738 differs between cell 206 lines used and, importantly, was independent of their HPV-status.
I recommend accepting the article after minor revision relating to the following points:
- Please, provide the rationale for using the given inhibitor concentrations and RT doses.
- I would recommend to change the order of the results presentation, i.e. I think that the results presented in Figure 2 should precede those presented in Fig. 1.
- I think that the term “clonogenic survival” used in the study requires more explanation. What do authors mean by “clonogenic” (survival or growth) in terms of cell lines? For eg. in the legend of Fig. 1 write that:
Clonogenic cell survival is shown as mean± s.e.m. [mean of what exactly?] relative to non-irradiated cells [and utreated with the inhibitor, I guess].
- The vehicle is described as just DSMO in cell line experiments while in animal model it was DMSO and propylene glycol. Is it correct?
- The results relating to H2AX are not commented in the Discussion section.
as well as some minor comments:
- Please, provide the meaning for all the abbreviations used in the abstract, as well as “DSBs” in section 2.3. of the manuscript.
- Reword the first sentence of the Introduction section: “…the alcohol and tobacco related and human papillomavirus…” (the first impression in that there are 3 types of in head and neck cancer.
- Reword the third sentence of the Introduction section: “many HNSCC patients face local tumor recurrences.” I feel that the word “face” is inappropriate, I would propose “experience” instead.
- In my opinion , the sentence “Dysfunction of the G1 cell cycle checkpoint, often via defects in the p53-Rb pathways, is one the major characteristics of cancer cells and results in reliance on the S and G2/M checkpoints [9].” In the Introduction section requires some explanation on its second part, i.e. the reliance on the S and G2/M checkpoints.
- In the legend of Fig. 4, the “indicated target proteins” text should be replaced with protein names.
Reviewer 2 Report
The authors investigated a combined action of ATR inhibitor AZD6738 and radiation on human papilloma virus-positive and negative head and neck squamous cell carcinoma cells. Three HPV+ and three HPV- cell lines were used. In addition, there are in vivo data obtained using mice that validated results obtained using the cells.
The manuscript is well-designed, well-structured, and nicely presented. There are several points that could further improve the presentation of the data.
- Kindly indicate molecular weights (as determined by ladders) next to the western blot images, Figure 4 and Figure 7.
- Section 2.3. Lines 117-132. While the title and the description of the results mention DNA damage and DNA repair, the authors only followed phosphorylation of histone H2AX. This assay is often and wrongly used to conclude on the DNA damage and DNA repair by itself. However, phosphorylation of H2AX only indirectly indicates, although usually correlates with, existing DSBs; it directly indicates an activated DDR. For more accurate presentation of the data, it is possible to discuss what exactly is measured by following the phosphorylated H2AX, and what is not. DSBs are not directly measured by this assay. Kindly check the title of section 2.3, whether "DNA damage" statement is appropriately used.
- Moreover, untreated cells ("control", Figure 5) possess relatively high levels of cells positive for phosphorylated H2AX (50-60%). What is the sensitivity of this assay? Is it possible to use the H2AX-deficient cells, or any other suitable negative control, to determine the real background of the assay? The manuscript might also benefit from discussing these points.
- Figure 6. It relies on n=2 (line 155), although the statistical analysis is performed and presented. For solid conclusion, n=3 or more. One option is to show a representative experiment and skip statistical analysis. Another option is to provide more data (n=3), which will allow concluding on the significance of the difference between the groups, if any.
- Materials and Methods. Kindly provide catalog numbers of the products used in the study and mentioned in this section.
Reviewer 3 Report
The authors investigate the ATR inhibitor AZD6783 and its synergism with RT in HPV+ and - HNSCC cell lines. They showed a reduction in clonal survival following RT+ ATRi treatment irrespective of HPV status. There was very little effect of ATRi by itself in the survival of the cells. The results are interesting but could be strengthened by a couple of experiments. In particular, an alternative way of assessing the number of gamma-H2AX cells would be good. It would also be good to show images of micronucleated cells and describe how the quantitation was carried out.
- In Figure 1, is the ATRi taken off after 2 hours or is it left on for 24 hours? What happens if the drug is kept on the cells throughout the clonogenic survival? It would be interesting to know this as the ATRi could be administered to a patient following RT. Repeated doses of ATRi could be used to maintain serum concentrations.
- In Figure 3, how do the authors know that the ATRi is not arresting the cells at different stages of the cell cycle? What is the timing of the drug treatment and radiation? If the drug was added first could it promote cell cycle arrest at various points in the cell cycle? So, rather than inhibiting RT G2/M arrest it masks this by prior arresting the cells?
- In Figure 5, representative gamma-H2AX blots should be shown, or provided as supplementary figures. Or, alternatively, simple staining of the cells. A background of 50% positivity in HPV negative cells seems very high. The fact that 6Gy does not increase this by much in the HPV negative lines is a result that requires to be confirmed by alternative means.
- In Figure 6, images of micronucleated cells should be provided. How many cells were counted? By what means? Double blind?
Round 2
Reviewer 2 Report
The authors modified the manuscript and addressed the reviewer's questions
Reviewer 3 Report
I thank the authors for responding reasonably to the requests made. This has helped clarify some points in the manuscript that the reader may want to know.